# Multicenter Retrospective Analysis of Second-Line Therapy after Gemcitabine Plus Nab-Paclitaxel in Advanced Pancreatic Cancer Patients

**DOI:** 10.3390/cancers12051131

**Published:** 2020-04-30

**Authors:** Valeria Merz, Alessandro Cavaliere, Carlo Messina, Massimiliano Salati, Camilla Zecchetto, Simona Casalino, Michele Milella, Orazio Caffo, Davide Melisi

**Affiliations:** 1Digestive Molecular Clinical Oncology Research Unit, University of Verona, 37134 Verona, Italy; alessandro.cav@hotmail.com (A.C.); camilla.zecchetto@gmail.com (C.Z.); simonacasalino@hotmail.it (S.C.); 2Department of Medical Oncology, Santa Chiara Hospital, 38122 Trento, Italy; carlo.messina@apss.tn.it (C.M.); orazio.caffo@apss.tn.it (O.C.); 3Section of Medical Oncology, University of Verona, 37134 Verona, Italy; michele.milella@univr.it; 4Department of Medical Oncology, University Hospital of Modena, 4121 Modena, Italy; maxsalati@live.it; 5PhD Program Clinical and Experimental Medicine (CEM), University of Modena and Reggio Emilia, 4121 Modena, Italy

**Keywords:** pancreatic cancer, second-line therapy, nal-IRI, nab-paclitaxel

## Abstract

Pancreatic cancer is one of the most lethal solid tumors. In many European countries gemcitabine plus nab-paclitaxel is the preferred first-line treatment. An increasing number of patients are eligible for second-line therapy, but the best regimen is still controversial. This study aimed to evaluate the efficacy of oxaliplatin-based compared to irinotecan-based therapies in this setting. 181 advanced pancreatic cancer patients consecutively treated in three centers with a second-line therapy progressed on gemcitabine plus nab-paclitaxel were retrospectively enrolled. OS and PFS were calculated using the Kaplan-Meier method and survival of the two groups was compared using the log-rank test. The median PFS and OS were respectively 3.5 (95%CI 3.2–3.8) and 8.8 months (95%CI 7.9–9.8) from second-line therapy in the overall population. The median PFS and OS were respectively 3.3 (95%CI 3.1–3.5) and 8.2 months (95%CI 7.24–9.34) with an irinotecan-based combination compared to 4.0 (95%CI 2.4–5.7) and 10.3 months (95%CI 8.62–12.02) in patients receiving an oxaliplatin-based combination. We observed a clear trend for longer survival outcomes with platinum-based doublet compared to regimens including irinotecan or nal-IRI. Head-to-head trials are still lacking. The neutrophil-to-lymphocyte ratio and the presence of liver metastases could drive physicians in tailoring the treatment strategy.

## 1. Introduction

Pancreatic cancer is one of the most lethal human malignancies, and the improvement of survival is one of the most urgent needs in cancer medicine [1,2]. Pancreatic cancer five-year relative survival rate remains at 7% and it is projected to become the second leading cause of cancer-related death by 2030 in Western countries [3,4]. The modest prolongation of pancreatic cancer patients’ survival in recent decades is in contrast to many other cancer types, in which screening, early detection, and novel treatments led to significant outcome improvements.

Pancreatic cancer is commonly diagnosed at a late stage. About 80% of patients are not eligible for surgery due to local vascular involvement or distant metastases.

In the last decade, two novel chemotherapeutic regimens have been registered for the first-line treatment of patients with metastatic disease. In the PRODIGE/ACCORD phase III trial, FOLFIRINOX (folinic acid, fluorouracil, irinotecan, oxaliplatin) significantly improved median overall survival (OS) compared to gemcitabine (11.1 months versus 6.8 months, hazard ratio, HR = 0.57, *p* < 0.001) [5]. In the MPACT phase III study, nab-paclitaxel plus gemcitabine significantly improved median OS compared to gemcitabine (8.5 months versus 6.7, HR 0.72, *p* < 0.001) [6]. Based on these results, however, the combination regimen of gemcitabine plus nab-paclitaxel remains the most frequent combination adopted in first-line metastatic setting in several European countries due to the safer toxicity profile if compared to FOLFIRINOX and the approval of nab-paclitaxel limited to the first-line treatment by national drug agencies [7]. Nonetheless, since they maintain a sufficient performance status (PS), about half of the patients failing a gemcitabine-based first-line regimen are commonly considered to receive second-line therapy [8,9].

To date, different chemotherapeutic regimens have been tested in this disease setting.

Phase III German CONKO (Charité Onkologie Clinical study group) trial compared OFF regimen (oxaliplatin, leucovorin, and 5-fluorouracil) with best supportive care (BSC). Although several methodological limitations are due to the unmet planned accrual, the OFF arm showed a statistically significant improvement in median OS of 4.82 versus 2.3 months over BSC (HR 0.45, 95% CI 0.24–0.83, *p* = 0.008) [10]. Consistently, in the phase III CONKO-003 trial OFF regimen was superior to FF (leucovorin and 5-fluorouracil) as the second-line treatment after progression under gemcitabine-containing regimens. This study showed a significant benefit for the combination arm over 5-fluorouracil (5-FU) and leucovorin (LV), reporting an improvement in PFS (2.9 versus 2.0 months, *p* = 0.019) and OS (5.9 versus 3.3 months, *p* = 0.01) [11]. Conversely, the PANCREOX study investigating a different oxaliplatin-containing regimen (mFOLFOX6) did not show any benefit compared to 5-FU/LV in progression free survival (PFS), which was 3.1 and 2.9 months, respectively [12]. Recently, results of the phase III SEQUOIA trial have been reported at the Gastrointestinal Cancers Symposium 2020, which evaluated the addition of pegilodecakin (PEG; pegylated IL-10) to FOLFOX in gemcitabine-refractory patients [13]. Median OS with FOLFOX + PEG was similar (5.8 months) to that of FOLFOX (6.3 months) with a HR of 1.045 (*p* = 0.66).

The randomized phase III NAPOLI-1 trial compared nanoliposomal irinotecan (nal-IRI) plus 5-FU/LV versus 5-FU/LV alone in advanced pancreatic cancer patients previously treated with gemcitabine-based chemotherapy. Nal-IRI plus 5-FU/LV demonstrated an OS advantage compared with 5-FU/LV (6.2 vs. 4.2 months, respectively, HR 0.75, 95% CI 0.57–0.99, *p* = 0.039) [14]. These findings led to United States FDA approval in October 2015 for nal-IRI in combination with 5-FU and LV to treat patients with metastatic pancreatic cancer previously treated with gemcitabine-based chemotherapy. One year later, this combination was also authorized by European Medicines Agency (EMA) in this clinical setting. Even though these results, nal-IRI has not received approval to reimbursement by the Italian Drug Agency (AIFA). 

Head-to-head clinical trials comparing nal-IRI- and oxaliplatin-based therapies in patients with gemcitabine-refractory metastatic pancreatic cancer are still lacking.

Here, we present survival outcomes with different second-line treatments in a real-world population of advanced pancreatic cancer patients previously treated with gemcitabine plus nab-paclitaxel.

## 2. Materials and Methods

### 2.1. Population

Patients affected by advanced pancreatic cancer and consecutively treated between January 2015 and December 2018 in three Italian centers with second-line combination therapy after failing the first-line treatment with gemcitabine plus nab-paclitaxel were included. Ethics approval and consequent informed consent were not required for this study, according to “Authorization n. 9/2016 –General Authorization for the Processing of Personal Data for Scientific Research Purposes–15 December 2016” (published in *Gazzetta Ufficiale* n. 303 of 29 December 2016). On the basis of this authorization, universities, research centers, and scientific societies do not require ethics approval to perform observational studies on data previously recorded without significant influence on affected patients.

Selection criteria included a cytological or histological confirmed diagnosis of advanced pancreatic adenocarcinoma; age ≥18 years; confirmed radiological disease progression after first-line treatment with gemcitabine + nab-paclitaxel regimen; treatment with second-line combination chemotherapeutic regimens for at least one cycle; availability of clinical and laboratory data at the beginning of second-line chemotherapy; subsequent availability of response evaluation and survival information. 

### 2.2. Evaluation of Outcomes

OS2 was defined as the timeframe from the beginning of second-line treatment to death from any cause or censored at the date of the latest follow-up. PFS2 was defined as the timeframe from second-line treatment to disease progression or death from any cause. Objective tumor response was evaluated according to the Response Evaluation Criteria in Solid Tumors (RECIST) ver. 1.1.

### 2.3. Statistical Analysis

The median OS and PFS were calculated using the Kaplan–Meier method. Survival duration was compared using the log-rank test. Fisher′s exact test was used for pairwise comparisons of objective response.

The effect of multiple factors on survival, the hazard ratio (HR) and its 95% confidence interval (CI) were evaluated using Cox proportional hazards models. All tests were two-sided and a *p* value < 0.05 was considered statistically significant. Statistical analyses were performed using SPSS 24.0 statistical software (SPSS, IBM Corporation, Somers, NY, USA).

## 3. Results

### 3.1. Population

We evaluated a consecutive series of 181 pancreatic cancer patients treated with second-line therapy previously progressed to gemcitabine and nab-paclitaxel (Figure 1).

Baseline patient characteristics before second-line treatment are summarized in Table 1.

The median age was 64 years (range 34–81 years) and there was a slight prevalence of males over females (respectively 54.6% versus 45.5%). Most patients had an ECOG PS of 0 (62.9%). Primary tumor sites were equally distributed in the head (50.9%) and body-tail (48.1%) of the pancreas. The majority of patients had a normal body mass index (BMI) (between 18.5 and 24.9 kg/m^2^). Most patients had increased levels of CA19.9 (78.5%). About half of the patients had two or more sites of metastasis (52.5%) and the liver was the most common secondary site (71.8%). About one-third (33.6%) of patients previously received a locoregional treatment and 16.6% of patients received neoadjuvant or adjuvant therapy.

### 3.2. Survival Outcomes and Response Rates of Second-Line Therapies in the Overall Population

The median PFS2 was 3.5 months (95% CI 3.2–3.8) and the median OS2 was 8.8 months (95%CI 7.9–9.8) from second-line therapy in the overall population (Figure 2). 

The objective response rate (ORR) in the overall population receiving second-line therapy was 10.5% (Table 2). Sixteen (9.3%) patients experienced a partial response (PR) and two (1.2%) a complete response (CR). Thirty-eight (22%) patients achieved disease stabilization as the best response. About one-third (32.3%) of patients achieved disease control.

Univariate analysis was performed on all available clinical factors. No difference in PFS and OS was detected examining gender, PS, primary tumor site, previous surgery or radiotherapy, liver and peritoneum metastases, BMI, CA19.9 levels, and neutrophil-to-lymphocyte ratio. Lung and bone metastases were significantly associated with better PFS at univariate analysis (Table 3). The presence of bone metastases was confirmed to be independently associated with OS at multivariate analysis.

### 3.3. Second-Line Therapies in the Overall Population

Fifty-two (28.7%) patients received an oxaliplatin-based doublet therapy, and in the majority of them (98.1%) the association included a fluoropyrimidine (34 patients received 5-FU, 17 capecitabine, 1 gemcitabine). One hundred and twenty (66.2%) patients have been treated with a combination of 5-FU and a topoisomerase I inhibitor. In particular, ninety-four received 5-FU and nal-IRI within an expanded access program and twenty-six received 5-FU plus irinotecan (FOLFIRI). Six (3.3%) patients received a triplet regimen with mFOLFIRINOX and three (1.6%) patients received capecitabine plus eribulin within a study protocol at the University Hospital of Modena.

### 3.4. Survival Outcomes and Response Rates in Patients Receiving Oxaliplatin-Based and Irinotecan-Based Doublet Regimens in the Second-Line Setting

To compare oxaliplatin-based with irinotecan-based doublet chemotherapeutic regimens, we excluded patients treated with triplet chemotherapy or with other drug regimens (Figure 1).

The median PFS2 was 3.3 months (95%CI 3.1–3.5) in patients receiving an irinotecan-based combination compared to 4.0 months (95%CI 2.4–5.7) in patients receiving an oxaliplatin-based combination (*p* = 0.494) (Figure 3). The median OS2 was 8.2 months (95%CI 7.24–9.34) compared to 10.3 months (95%CI 8.62–12.02), respectively (*p* = 0.713) (Figure 3).

The disease control rate (DCR) was 35.2% in patients receiving an oxaliplatin-based therapy compared to 30.7% in patients receiving an irinotecan-based doublet (*p* = 0.593) (Table 2). The ORR was 13.7% in patients receiving an oxaliplatin-based therapy compared to 9.6% in patients treated with an irinotecan-based doublet (*p* = 0.433).

At the univariate analysis, no clinical features revealed a statistically significant advantage in PFS2 or OS2 for patients receiving an oxaliplatin-based regimen compared to an irinotecan-based regimen (Table 4).

Forty-seven (26.2%) patients experienced a grade 3 or 4 toxicity. In the oxaliplatin-based arm 41.4% of patients developed a grade 3 or 4 toxicity compared to 20.3% in the irinotecan-based group of patients.

### 3.5. Survival Outcomes of First-Line Therapy in the Overall Population

The median PFS of the first-line therapy with gemcitabine plus nab-paclitaxel was 6.4 months (95%CI 6.0–6.9) and the median OS from the start of the first-line therapy was 17.2 months (95%CI 15.6–18.8) in the overall population.

## 4. Discussion

Most patients affected by advanced pancreatic cancer invariably progress within few months during or after first-line therapy. In the last decade, the clinical scenario where most patients received just a single line of therapy because of the lack of effective therapeutic options dramatically changed with more than half of the patients being currently eligible for second-line chemotherapy [8].

In this setting, the best therapeutic sequence is still a matter of debate. An increasing number of clinical trials are exploring the efficacy of second-line therapy for patients especially in those treated with first-line gemcitabine-based therapies [9].

To date, NAPOLI-1 study remains the largest phase III randomized trial in this setting establishing nal-IRI plus 5-FU/LV as a novel standard of care [15,16,17]. Only small retrospective studies support the use of standard irinotecan [18] and FOLFIRI regimen remains an option in countries where nal-IRI is not reimbursed.

Conflicting results on the use of oxaliplatin are reported [11,12]. Even if the use of FOLFOX is not supported by robust evidence, it might be an option in the routine daily practice, and is thus used as a control regimen in some randomized clinical trials [13].

Finally, only a very limited percentage of patients in the routine clinical practice receive a triplet therapy in second-line setting.

No prospective trials comparing the two doublet regimens after first-line therapy with gemcitabine plus nab-paclitaxel have been conducted until now.

In the present study, we report a real-world experience of three referral centers for advanced pancreatic cancer patients receiving a second-line therapy previously progressed on gemcitabine plus nab-paclitaxel. In our series, the PFS observed with gemcitabine plus nab-paclitaxel was longer than that reported in the pivotal MPACT trial [6]. Similarly, the magnitude of benefit from a second-line therapy seems to be even greater than what previously reported, achieving a mPFS of 3.5 months and a mOS of 8.8 months. ORR and DCR were 10.5% and 32.3% respectively in the overall population.

Interestingly, about two-thirds of patients in our series received a doublet regimen including 5-FU and topoisomerase I inhibitors. This could find two possible explanations. Firstly, the frequent residual nab-paclitaxel-related sensory neuropathy could lead physicians to avoid a neurotoxic agent as oxaliplatin. Then, at the time of patient enrollment, a program of compassionate use of nal-IRI in Italy and other European countries was active. Thus, nal-IRI was offered to many patients.

In this study, we attempted to measure the clinical outcomes obtained with chemotherapeutic regimens including a platinum salt in comparison with those including standard or liposomal topoisomerase inhibitors. No statistically significant differences were found in terms of survival or response rate between these treatment classes. However, there is a signal in favor of oxaliplatin-based compared to irinotecan-based combinations and the lack of statistical significance of this trend could be explained with the small size of the two groups of patients. At the univariate analysis, the clinical characteristics that were correlated to a greater benefit from oxaliplatin-based therapy included a PS 0, a pancreatic head tumor, the presence of liver metastases and a neutrophil-to-lymphocyte ratio higher than 5. These findings gain even more relevance when compared to the results of NAPOLI-1 trial. Wang-Gillam et al. reported, indeed, a lower neutrophil-to-lymphocyte ratio and absence of liver metastases as positive predictive factors of response to nal-IRI+5-FU/LV [15]. A different pattern of metastatization could reflect different biology and could explain a diverse response to drugs.

It is of primary importance to identify novel biomarkers that could drive treatment choices. In pancreatic cancer, the first biomarker-driven approach was recently established in the POLO trial in patients carrying germline Breast Related Cancer Antigens (BRCA) gene mutations [19]. Mutations in BRCA1/2 and in Partner and Localizer of *BRCA2* (PALB2) genes are reported in approximately 5% to 9% of pancreatic cancer patients. Patients with germline BRCA mutations and other defects of homologous recombination repair of DNA are characterized by an increased sensitivity to platinum agents and poly(ADP-ribose) polymerase (PARP) inhibitors [20]. The identification of mutations involving DNA damage repair pathway, also as somatic events, could guide the treatment strategy also in the second-line setting [21,22].

We acknowledge some limitations of our study. The major limit is the retrospective design. However, the accurate and rigid selection of patients might have limited this bias. We are conscious of the small sample size limiting the value of subgroup analysis. However, we included 181 patients in a specific setting, making this series numerically consistent. Most of the previous studies included patients receiving gemcitabine-based therapies in the first-line, whereas we selected exclusively patients treated with gemcitabine and nab-paclitaxel. The inclusion of a novel chemotherapeutic drug as nal-IRI is another strength of our study.

## 5. Conclusions

Because of the conflicting results about the oxaliplatin-containing regimens, nal-IRI remains the most novel and solid option as second-line treatment according to ESMO [23] and NCCN guidelines [24]. However, in different European countries, nal-IRI is currently not reimbursed and in the clinical practice irinotecan- and oxaliplatin-based combination regimens continue to be routinely used. The choice between irinotecan- and oxaliplatin-based chemotherapy should be driven by PS, residual toxicities from prior treatments, response to prior therapies, disease burden and patient preference.

Our present study did not show a statistically significant difference in survival duration in advanced pancreatic cancer patients receiving irinotecan- or oxaliplatin-based chemotherapy as second-line after failing gemcitabine plus nab-paclitaxel chemotherapy. We observed a signal for longer survival outcomes with platinum-based doublet compared to regimens including irinotecan or nal-IRI. However, the retrospective design and the limited number of patients might mitigate these conclusions.

In the absence of comparative trials, potential predictive factors are needed. Data from our series suggest that a neutrophil-to-lymphocyte ratio higher than 5 and the presence of liver metastases could predict a better outcome from oxaliplatin-based therapies if compared to the irinotecan-based ones.

Prospective, randomized trials are needed in this specific setting. Our study suggests neutrophil-to-lymphocyte ratio and the presence of liver metastases as potential selection factors for the second-line therapy in pancreatic cancer patients failing first-line gemcitabine plus nab-paclitaxel regimen.

## Figures and Tables

**Figure 1 cancers-12-01131-f001:**
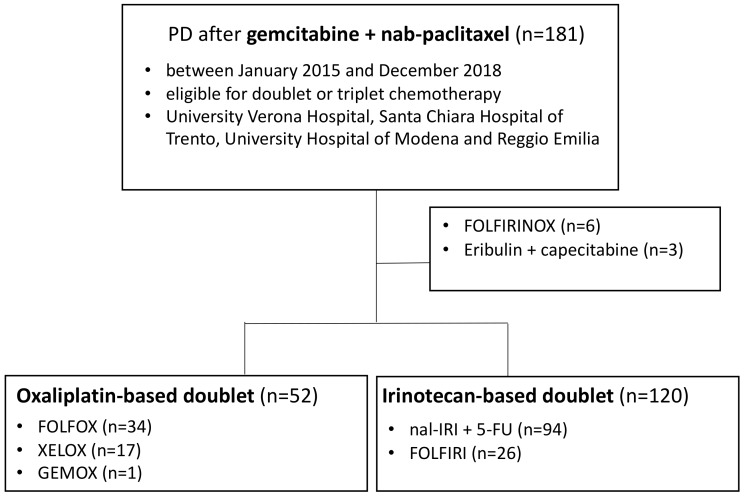
Selection of patients included in the study.

**Figure 2 cancers-12-01131-f002:**
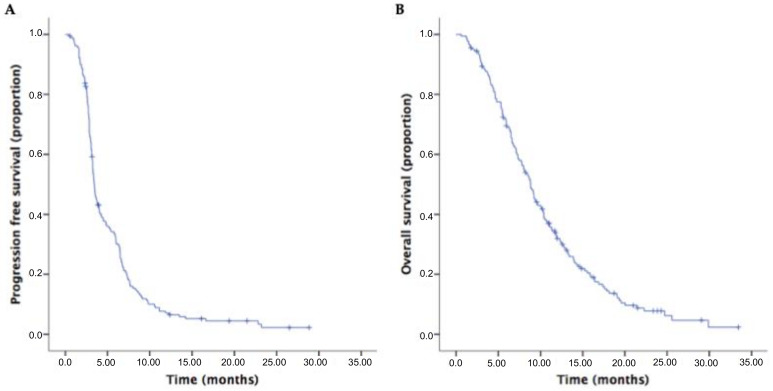
Kaplan–Meier curves of progression-free survival (PFS) (**A**) and of overall survival (OS) (**B**) in the overall second-line population.

**Figure 3 cancers-12-01131-f003:**
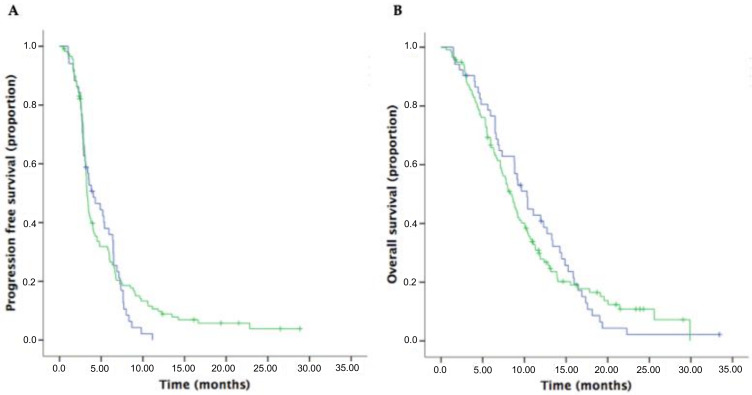
Kaplan–Meier curves of PFS (**A**) and OS (**B**) in the oxaliplatin- (blue lines) and irinotecan-based (green lines) regimen cohorts.

**Table 1 cancers-12-01131-t001:** Baseline clinical characteristics of overall second-line population.

Baseline Characteristics	*n* (%)
Sex	
female	82 (45.4)
male	99 (54.6)
Median age (years)	64 (34-81)
ECOG PS	
0	114 (62.9)
1	49 (27.1)
2	11 (6.0)
unknown	7 (3.9)
Tumor location	
head	92 (50.9)
body-tail	87 (48.1)
unknown	2 (1.2)
CA19.9	
<40 U/mL	24 (13.2)
≥40 U/mL	142 (78.5)
unknown	15 (8.3)
BMI	
≥25	42 (23.2)
18.5–24.9	113 (62.5)
<18.5	13 (7.2)
unknown	13 (7.2)
Metastatic sites	
liver	130 (71.8)
peritoneum	43 (23.8)
lung	59 (32.6)
other	16 (8.8)
Number of metastatic sites	
0	3 (1.6)
1	83 (45.9)
2	59 (32.6)
≥3	36 (19.9)
Previous procedures	
radiotherapy	19 (10.4)
surgery	42 (23.2)
Prior neoadjuvant therapy	14 (7.7)
Prior adjuvant therapy	16 (8.8)

**Table 2 cancers-12-01131-t002:** Overall response rate (ORR) and disease control rate (DCR) in the overall population and in the oxaliplatin- and irinotecan-based doublet.

Response	Overall Population *n* (%)	Oxaliplatin-Based Doublet *n* (%)	Irinotecan-Based Doublet *n* (%)
**ORR**	18 (10.5)	7 (13.7)	11 (9.6)
**PR**	16 (9.3)	7 (13.7)	9 (7.8)
**CR**	2 (1.2)	0	2 (1.7)
**DCR**	56 (32.3)	18 (35.2)	35 (30.7)
**SD**	38 (22.0)	11 (21.5)	24 (21.1)
**PR**	16 (9.3)	7 (13.7)	9 (7.8)
**CR**	2 (1.2)	0	2 (1.7)

**Table 3 cancers-12-01131-t003:** Univariate and multivariate analyses of clinical characteristics influencing progression free survival (PFS) and overall survival (OS) in the overall second-line population.

Variables	PFS		OS	
	Univariate	Multivariate	Univariate	Multivariate
	*p*	*p*	*p*	*p*
Sex				
Male vs. female	0.935		0.274	
PS				
0 vs. 1–2	0.062		0.077	
Pancreatic tumor site				
Head vs. body-tail	0.639		0.318	
Previous surgery				
No vs. yes	0.304		0.126	
Previous radiotherapy				
No vs. yes	0.213		0.901	
Site of metastasis				
Liver (no vs. yes)	0.865		0.051	
Lung (no vs. yes)	**0.048**	0.060	0.624	
Peritoneum (no vs. yes)	0.123		0.237	
Bone (no vs. yes)	**0.002**	**0.004**	0.070	
BMI				
≤25 vs. >25 kg/m^2^	0.099		0.058	
CA19.9				
<40 vs. ≥40 U/mL	0.069		0.172	
N/L ratio				
≤5 vs. >5	0.770		0.809	

Bold numbers: statistically significant, further evaluated at the multivariate analysis.

**Table 4 cancers-12-01131-t004:** Comparison in terms of PFS and OS between oxaliplatin- and irinotecan-based doublet cohorts for every clinical characteristic at baseline.

PFS				OS		
Characteristic	Oxaliplatin-Based Doublet Months (95%CI)	Irinotecan-Based Doublet Months (95%CI)	*p*	Oxaliplatin-Based Doublet Months (95%CI)	Irinotecan-Based Doublet Months (95%CI)	*p*
Sex						
Male	4.0 (2.2–6.0)	3.2 (3.1–3.5)	0.446	8.8 (8.0–9.6)	8.0 (5.3–10.7)	0.972
Female	3.5 (0.6–6.5)	3.3 (3.0–3.7)	0.901	11.9 (7.8–16)	8.5 (6.9–10.1)	0.550
PS						
0	4.0 (1.4–6.7)	3.6 (2.8–4.5)	0.323	11.0 (8.1–14.0)	9.1 (7.3–10.9)	0.843
1–2	2.8 (1.7–4.1)	2.8 (2.5–3.2)	0.871	6.8 (5.2–8.5)	5.9 (3.6–8.3)	0.905
Pancreatic tumor site						
Head	5.3 (3.9–6.9)	3.3 (2.9–3.6)	0.532	10.3 (6.3–14.3)	8.0 (6.3–9.7)	0.211
Body-tail	3.1 (2.2–4.1)	3.5 (3.0–4.0)	0.085	9.1 (7.2–11.1)	8.9 (1.2–6.7)	0.355
Previous surgery						
Yes	5.1 (1.1–9.2)	3.7 (2.6–4.9)	0.450	10.3 (7.7–13.1)	10.0 (6.9–13.2)	0.621
No	3.8 (2.6–4.9)	3.2 (3.0–3.5)	0.855	9.2 (7.4–11.1)	7.7 (6.2–9.3)	0.417
Previous radiotherapy						
Yes	7.1 (3.2–11.1)	4.5 (1.1–7.8)	0.813	12.7 (4.2–21.2)	9.1 (7.9–10.6)	0.679
No	3.5 (2.6–4.4)	3.2 (3.1–3.4)	0.428	9.2 (6.4–11.9)	8.0 (6.6–9.4)	0.789
Site of metastasis						
Liver	3.5 (2.5–4.4)	3.3 (2.9–3.9)	0.361	9.2 (7.5–10.9)	7.7 (6.6–8.9)	0.831
Lung	3.0 (2.5–3.6)	3.0 (2.6–3.5)	0.903	8.8 (4.8–12.9)	9.3 (7.1–11.6)	0.889
Peritoneum	2.8 (2.3–3.5)	2.9 (2.6–3.4)	0.554	6.6 (5.0–8.2)	7.7 (4.7–10.9)	0.676
Bone	1.7 (-)	2.1 (1.0–3.4)	0.321	2.6 (-)	7.7 (2.2–13.4)	0.145
BMI						
≤25 kg/m^2^	3.5 (1.9–5.1)	3.2 (2.9–3.6)	0.751	9.1 (5.8–12.5)	8.0 (6.8–9.2)	0.792
>25 kg/m^2^	4.0 (1.8–6.3)	3.7 (0.6–7.0)	0.415	10.3 (4.1–16.7)	10.0 (7.0–13.1)	0.639
CA19.9						
<40 U/mL	6.5 (5.5–7.5)	4.0 (2.8–5.2)	0.857	14.8 (13.0–16.7)	7.4 (4.9–9.9)	0.959
≥40 U/mL	3.5 (2.4–4.6)	3.2 (3.0–3.5)	0.482	9.6 (8.0–11.3)	8.6 (7.4–9.9)	0.827
N/L ratio						
≤5	3.8 (2.6–5.1)	3.3 (3.2–3.6)	0.364	10.3 (7.8–12.9)	8.0 (6.4–9.7)	0.564
>5	6.4 (6.3–6.6)	2.8 (2.6–3.2)	0.384	12.2 (7.0–17.5)	7.7 (3.6–11.8)	0.816

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
