# Peer review of "Multicenter Retrospective Analysis of Second-Line Therapy after Gemcitabine Plus Nab-Paclitaxel in Advanced Pancreatic Cancer Patients"

_cancers, 2020, doi:10.3390/cancers12051131_

Round 1

Reviewer 1 Report

Pancreatic cancer is one of the most lethal solid tumour. 

The authors aimed to show the efficacy of different drug combinations.

There are few spelling mistakes and text editing but a part from that I am happy with the conclusions.  

below my "minor" comments:

line 43: reference [11]. space (missing) line 49: . end of sentence lines 111 and 112: number of patients expressed using Arabic numbers whereas lines 124, 126, 127-130 and 152: numbers are expressed using letters Fig.3: Fig3B is overlapping with Fig3A so that the months scale is not fully visible Table 4: line up title (PFS and OS) with below content lines 171, 181, 220: different references style line 206: double space "DNA are" line 207: no space inhibitors[ref]

Regards

Author Response

Pancreatic cancer is one of the most lethal solid tumour. 

The authors aimed to show the efficacy of different drug combinations.

There are few spelling mistakes and text editing but a part from that I am happy with the conclusions.  

below my "minor" comments:

  1. line 43: reference [11]. space (missing)

 Done.

  1. line 49: . end of sentence

Done

  1. lines 111 and 112: number of patients expressed using Arabic numbers whereas lines 124, 126, 127-130 and 152: numbers are expressed using letters

Thank you for this observation. We modified the text by expressing numbers only with letters.

  1. 3: Fig3B is overlapping with Fig3A so that the months scale is not fully visible

We modified figures 2 and 3 to make them more clear and accurate.

  1. Table 4: line up title (PFS and OS) with below content

Thank you for your comment. Done.

  1. lines 171, 181, 220: different references style

We modified the references style in these lines adapting them to the rest of the paper.

  1. line 206: double space "DNA are"

We removed one space.

  1. line 207: no space inhibitors[ref]

We added one space.

Reviewer 2 Report

The authors conducted a multicenter retrospective study of second-line chemotherapy after gemcitabine plus nab-paclitaxel in advanced pancreastic cancer patients that compared oxaliplatin-based doublet (n=52) with irinotecan-based doublet (n=120). No statistically differences were found in terms of OS, PFS, and RR between the two treatment classes.

The study was well done and manuscript was well written.

I have some comments.

It is difficult to recommend oxaliplatin-based doublet in patients with a neutron-to-lymphocyte ratio higher than 5, presence of liver metastases, because there was no significantly difference between the two treatment classes. And this is a retrospective study with a relatively small numbers of patients.

In clinical setting, nab-paclitaxel-related sensory neuropathy is occurred often. So, physicians do not want to use oxaliplatin. As the study showed no benefit in oxaliplatin-based doublet over irinotecan-based doublet, I think no reason to choice oxaliplatin-based doublet.

On page 2, line 58, “EMA” should be spelled out.

In Figure 2, please provide colors of line in the figure legend.

Author Response

The authors conducted a multicenter retrospective study of second-line chemotherapy after gemcitabine plus nab-paclitaxel in advanced pancreatic cancer patients that compared oxaliplatin-based doublet (n=52) with irinotecan-based doublet (n=120). No statistically differences were found in terms of OS, PFS, and RR between the two treatment classes.

The study was well done and manuscript was well written.

I have some comments.

  1. It is difficult to recommend oxaliplatin-based doublet in patients with a neutron-to-lymphocyte ratio higher than 5, presence of liver metastases, because there was no significantly difference between the two treatment classes. And this is a retrospective study with a relatively small numbers of patients.

Thank for your comment. As already acknowledged in the discussion of the manuscript, our conclusions cannot be actually translated in the clinical practice because these results are not statistically significant, maybe also for the relative small number of patients. However, our results go somehow in the same direction of those of the NAPOLI-1 trial and, furthermore, suggest that possible negative predictive factors for irinotecan-based therapy could be on the other hand positive predictive factors for oxaliplatin-based therapies. This hypothesis will need to be further evaluated in prospective randomized trials.

  1. In clinical setting, nab-paclitaxel-related sensory neuropathy is occurred often. So, physicians do not want to use oxaliplatin. As the study showed no benefit in oxaliplatin-based doublet over irinotecan-based doublet, I think no reason to choice oxaliplatin-based doublet.

Thank you for this comment. In our analysis there is a not significant trend for a greater efficacy of oxaliplatin-based therapies compared to regimens including irinotecan. Our conclusions are not aimed at proposing the use of oxaliplatin after gemcitabine plus nab-paclitaxel in all pancreatic cancer patients. On the contrary, we think that the strongest prospective randomized data support the use of nal-IRI plus 5-FU. Furthermore, as you suggested, neurotoxicity often preclude the use of oxaliplatin after nab-paclitaxel. However, our data suggest that maybe a portion of patients could benefit from oxaliplatin-based regimens more than from irinotecan-based regimens and the use of predictive factors could contribute to select these patients and individualize therapy in this setting.

  1. On page 2, line 58, “EMA” should be spelled out.

We spelled out EMA.

  1. In Figure 2, please provide colors of line in the figure legend.

Thank you for your suggestion. We specified in the figure legend that oxaliplatin-based regimens are indicated with blue lines and irinotecan-based regimens with green lines.